# Symptomatic malaria enhances protection from reinfection with homologous *Plasmodium falciparum* parasites

**Christine F. Markwalter**[1☉], **Jens E. V. Petersen**[2☉], **Erica E. Zeno**[2,3], **Kelsey M. Sumner**[2,3], **Elizabeth Freedman**[2], **Judith N. Mangeni**[4], **Lucy Abel**[5], **Andrew A. Obala**[6], **Wendy Prudhomme-O'Meara**[1,2,4], **Steve M. Taylor**[1,2,3]*

**1** Duke Global Health Institute, Duke University, Durham, North Carolina, United States of America, **2** Division of Infectious Diseases, School of Medicine, Duke University, Durham, North Carolina, United States of America, **3** Department of Epidemiology, Gillings School of Global Public Health, University of North Carolina, Chapel Hill, North Carolina, United States of America, **4** School of Public Health, College of Health Sciences, Moi University, Eldoret, Kenya, **5** Academic Model Providing Access to Healthcare, Moi Teaching and Referral Hospital, Eldoret, Kenya, **6** School of Medicine, College of Health Sciences, Moi University, Eldoret, Kenya

☉ These authors contributed equally to this work.

* steve.taylor@duke.edu

**Data Availability Statement:** Parasite sequence data are available from NCBI (PRJNA646940). Data and code are available on Github (https://github.

## Abstract

A signature remains elusive of naturally-acquired immunity against *Plasmodium falciparum*. We identified *P. falciparum* in a 14-month cohort of 239 people in Kenya, genotyped at immunogenic parasite targets expressed in the pre-erythrocytic (circumsporozoite protein, CSP) and blood (apical membrane antigen 1, AMA-1) stages, and classified into epitope type based on variants in the DV10, Th2R, and Th3R epitopes in CSP and the c1L region of AMA-1. Compared to asymptomatic index infections, symptomatic malaria was associated with reduced reinfection by parasites bearing homologous CSP-Th2R (adjusted hazard ratio [aHR]:0.63; 95% CI:0.45–0.89; p = 0.008) CSP-Th3R (aHR:0.71; 95% CI:0.52–0.97; p = 0.033), and AMA-1 c1L (aHR:0.63; 95% CI:0.43–0.94; p = 0.022) epitope types. The association of symptomatic malaria with reduced hazard of homologous reinfection was strongest for rare epitope types. Symptomatic malaria provides more durable protection against reinfection with parasites bearing homologous epitope types. The phenotype represents a legible molecular epidemiologic signature of naturally-acquired immunity by which to identify new antigen targets.

## Author summary

The targets and mechanisms of naturally-acquired immune responses to *P. falciparum* parasites remain obscured, owing in part to the absence of a legible signature of immunity in humans exposed to repeated natural infections. A hallmark of functional protection for malaria and other pathogens is a reduced risk of re-infection with homologous strains. In a community-based, longitudinal cohort in a high transmission setting in Western Kenya, we analyzed by deep sequencing immunogenic segments of parasite antigens and the time

com/duke-malaria-collaboratory/mozzie_epitope_
analysis.git).

**Funding:** This work was supported by the National Institute of Allergy and Infectious Diseases [R21AI126024 to W.P-O. and R01AI146849 to W.P-O. and S.M.T.]. J.E.V.P. was supported by the Alfred Benzon Foundation, and C.F.M. was supported by F32AI149950. The funders had no role in study design, data collection and analysis, decision to publish, or preparation of the manuscript.

**Competing interests:** The authors have declared that no competing interests exist.

to reinfection following parasite clearance. When classifying parasites at described epitopes within targets of antiparasite immunity, we observed that, compared to parasites cleared following an asymptomatic infection, the hazard of reinfection following a symptomatic infection was reduced by 30–40%. The association of symptomatic infection with delayed homologous reinfection offers a signature of protection by which to identify novel targets of antiparasite immunity.

## Introduction

Malaria kills over 400,000 people annually [1] owing in part to the slow development of incomplete, naturally-acquired immunity (NAI). NAI provides reliable protection against severe malaria after a few infections [2] but does not wholly prevent symptomatic malaria, owing in part to short-lived immune responses and to *P. falciparum* antigenic variation that collectively limit the effectiveness of antibodies. [3–7] The mechanisms and targets of functional responses remain incompletely understood, complicating efforts to recapitulate these responses for the purposes of vaccine development. The rational design of the next generation of *P. falciparum* vaccines that are informed by mechanistic insights furnished by models of NAI would be enhanced by the identification of robust phenotypes of acquired protection in naturally-exposed populations.

Similar to other pathogens, a hallmark of functional protection against malaria is the delayed acquisition or decreased risk of infection when exposed to homologous parasites. [8] The phenotype of homologous protection against either infection or severity has been observed using a variety of approaches including challenge studies of a radiation-attenuated *Plasmodium falciparum* sporozoite vaccine [9,10] and of *P. vivax* used as neurosyphilis therapy,[11] as well as in field studies of monovalent vaccines targeting *P. falciparum* apical membrane antigen 1 (AMA-1) [12] and circumsporozoite protein (CSP). [7] Detecting this signature of functional protection is a challenge in observational field studies because the frequency of infections renders such studies operationally complex, and the genetic diversity of parasites and multiplicity of infections complicates strain-typing methods. Capturing homologous reinfection events and measuring associations with host and parasite factors in a natural setting can provide insights into the process by which anti-parasite immune memory is inscribed.

Herein, we explore how exposure to diverse sequence types in a high-transmission setting influences the time to reinfection with *P. falciparum*. We focused on immunogenic segments of antigens expressed in the pre-erythrocytic (CSP) and the blood (AMA-1) stages in our previously-published *P. falciparum pfcsp* and *pfama1* sequences collected in our longitudinal cohort of 239 people who suffered 902 asymptomatic and 137 symptomatic *P. falciparum* infections over 14 months of observation. [13] Because features of immune responses to *P. falciparum* differ between symptomatic malaria, which is associated with overexpression of pro-inflammatory cytokines IFN-γ, TNF, and IL-1β, [14] and asymptomatic infections, we explored if symptomaticity influenced the risk of reinfection with parasites bearing homologous CSP and AMA-1 epitope types. Given that symptomatic infection is associated with the subsequent development of more robust functional immune response for other pathogens such as SARS-CoV-2, [15] we hypothesized that, compared to asymptomatic infections, symptomatic malaria would prolong the time to reinfection with parasites bearing homologous CSP and AMA-1 types.

## Results

### Analytic population and risk of reinfection

As previously reported, between June 2017 and July 2018 in Webuye, Kenya, we collected 902 asymptomatic and 137 symptomatic *P. falciparum* infections; from these, amplification and sequencing was successful for *pfcsp* in 861 samples and for *pfama* in 724 samples. [13] For each participant, we defined the start and end of infection episode, each composed of a single or of multiple contiguous *P. falciparum* positive samples (Fig 1) and then calculated the time to reinfection as the time from the end of one index episode to the start of the next. We excluded from analysis index episodes for which time to reinfection was less than or equal to 60 days for 3 reasons: 1) our analysis was motivated to detect a signature of functional immunity which would be unlikely to manifest immediately; 2) to minimize the risk of bias introduced by either recrudescent infections after symptomatic infection or falsely-negative haplotype capture after asymptomatic infection; and 3) our sampling scheme of monthly active surveillance for asymptomatic infections, when combined with the definition of an end of infection as the absence of the parasite or type in the next sampling, resulted in an effective floor of 60 days for reinfection following an asymptomatic infection that was not present for passively-detected symptomatic infections. These excluded index episodes were more likely to be symptomatic (**Table A in** S1 Text) and to occur in the high transmission season, but were otherwise similar to the non-excluded infections. Therefore, we computed the time to reinfection following 275 asymptomatic and 69 symptomatic infection episodes that composed the analytic dataset. Between asymptomatic and symptomatic infections, we observed differences in transmission season, multiplicity of infection (MOI), parasite density, and read coverage. However, read coverage was high in both the asymptomatic and symptomatic groups, so it is unlikely that read coverage is driving MOI (Table 1).

We first compared between symptomatic and asymptomatic episodes the risk of reinfection among all participants irrespective of parasite sequence type. The median time to reinfection was 126 days (95% CI 119–150), and the probability of remaining free of reinfection within 1 year of index exposure was 0.07 (95% CI 0.04–0.13) (Fig 2A). In a multivariate mixed effects Cox proportional hazards model, compared to asymptomatic infections, the presence of symptoms during the index infection was not associated with the hazard of reinfection (aHR 1.04; 95% CI 0.74–1.46) (Fig 2B).

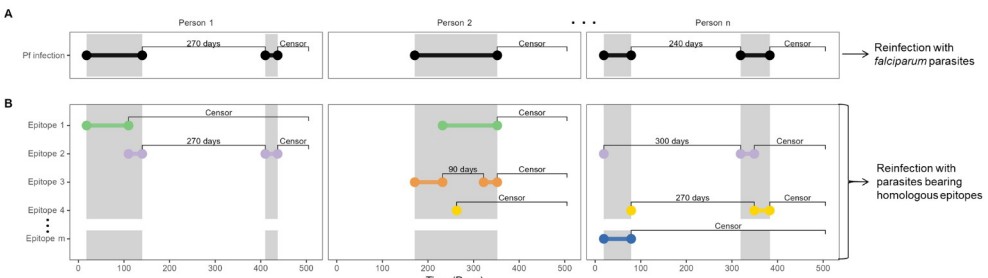

**Fig 1. Calculation of time-to-reinfection based on infection episodes and on type-specific episodes.** Illustration using mock data of the calculation of time-to-reinfection for 3 participants with multiple episodes. Dots indicate beginning and end of episodes, which are joined by horizontal bars. Gray shaded area indicates the time period of continuous *P. falciparum* positivity. Labels between bars indicate either time-to-reinfection (in days) or censoring following the end of an episode. (A) Pf infection indicates episodes of infection with *P. falciparum*. (B) Epitope X indicate episodes of infection with specific epitope types of *P. falciparum*, which are classified on the basis of translated nucleotide sequences of *csp* and *ama1*. Colors indicate unique CSP or AMA-1 epitopes which comprise multi-clonal infections of *P falciparum*.

**Table 1. Comparison of symptomatic and asymptomatic index infection episodes.**

| | Total | Asymptomatic (n = 275) | Symptomatic (n = 69) | p-value* |
|---|---|---|---|---|
| **Age, no. (%)** | | | | **0.003** |
| < 5y | 52 (15) | 43 (16) | 9 (13) | |
| 5–15 | 129 (38) | 91 (33) | 38 (55) | |
| > 15y | 162 (47) | 140 (51) | 22 (32) | |
| **Sex, no. (%)** | | | | **0.3** |
| Female | 194 (56) | 159 (58) | 35 (51) | |
| Male | 150 (44) | 116 (42) | 34 (49) | |
| **Transmission season, no. (%)\*\*** | | | | **< 0.001** |
| High | 148 (43) | 102 (37) | 46 (67) | |
| Low | 196 (57) | 173 (63) | 23 (33) | |
| **Multiplicity of infection, no. (%)\*\*\*** | | | | **< 0.001** |
| 1 | 100 (35) | 69 (31) | 31 (48) | |
| 2–5 | 100 (35) | 74 (34) | 26 (41) | |
| > 5 | 84 (30) | 77 (35) | 7 (11) | |
| **Parasite density, parasites/μL, median (IQR)** | **2.42 (0.47–62.79)** | **1.11 (0.40–9.62)** | **637 (50.0–4,770)** | **< 0.001** |
| **Read coverage, median (IQR)** | | | | |
| *pfcsp* | 6,719 (1,957–13,206) | 4,679 (1,526–11,791) | 12,380 (7,312–16,136) | < 0.001 |
| *pfama1* | 7,432 (3,200–12,939) | 6,318 (2,720–12,014) | 10,380 (7,015–14,880) | < 0.001 |

μL: microliter; *pfcsp*: *Plasmodium falciparum* circumsporozoite protein; *pfama1*: *P. falciparum* apical membrane antigen 1

* Computed by either the chi-square test or Wilcoxon rank sum test.

** Classified on the basis of the abundance of mosquitos collected in the prior two weeks into high (> 50) or low (≤ 50).

*** Defined as the larger of the number of unique *pfcsp* or *pfama1* nucleotide sequences in the last positive sample of an episode.

## *P. falciparum* CSP and AMA-1 amino acid diversity

To enable an analysis of reinfection with homologous parasites, we next assessed the parasite genotypes within these infections, in which, as previously reported, [13] we observed 155 unique *pfcsp* and 209 unique *pfama* nucleotide sequences. These encoded 145 unique CSP amino acid (AA) sequences with missense substitutions, which resulted from variance in 39 of the 95 AA positions captured by sequencing (Fig 3A), and 203 unique AMA-1 AA sequences, which resulted from variance in 50 of the 99 AA positions captured by sequencing (Fig 3C).

In order to reduce these highly diverse antigen segments into broader, biologically relevant groups and preserve statistical power, we identified the (1) most informative variable AA positions that (2) occurred in immunologically-relevant antigen segments. To focus on AA positions that would be most informative, we utilized a Random Forest approach to rank the ability of the variant AA positions to correctly predict the full nucleotide sequences (Fig 3B). The most informative AAs were clustered in three main variable areas of our CSP sequence: in the linker to the NANP repeat region harboring the DV10 epitope, the Th2R CD4+ T-cell epitope, [16] and the Th3R CD8+ T-cell epitope. [17] For AMA-1 the most informative among the 50 variant positions AA positions were distributed across four apparent clusters, one of which included the immunogenic c1L domain [18] (Fig 3D).

Next, we classified each parasite in each infection based on the four most predictive AA positions in the i) CSP DV10 epitope, ii) CSP Th2R epitope, iii) CSP Th3R epitope, and iv) AMA-1 c1L domain (hereafter CSP-DV10, CSP-Th2R, CSP-Th3R, or AMA-1 c1L epitope types, respectively). We observed 8 unique CSP-DV10 epitope types, 27 unique CSP-Th2R epitope types, 14 unique CSP-Th3R epitope types, and 20 unique AMA-1 c1L types. Within

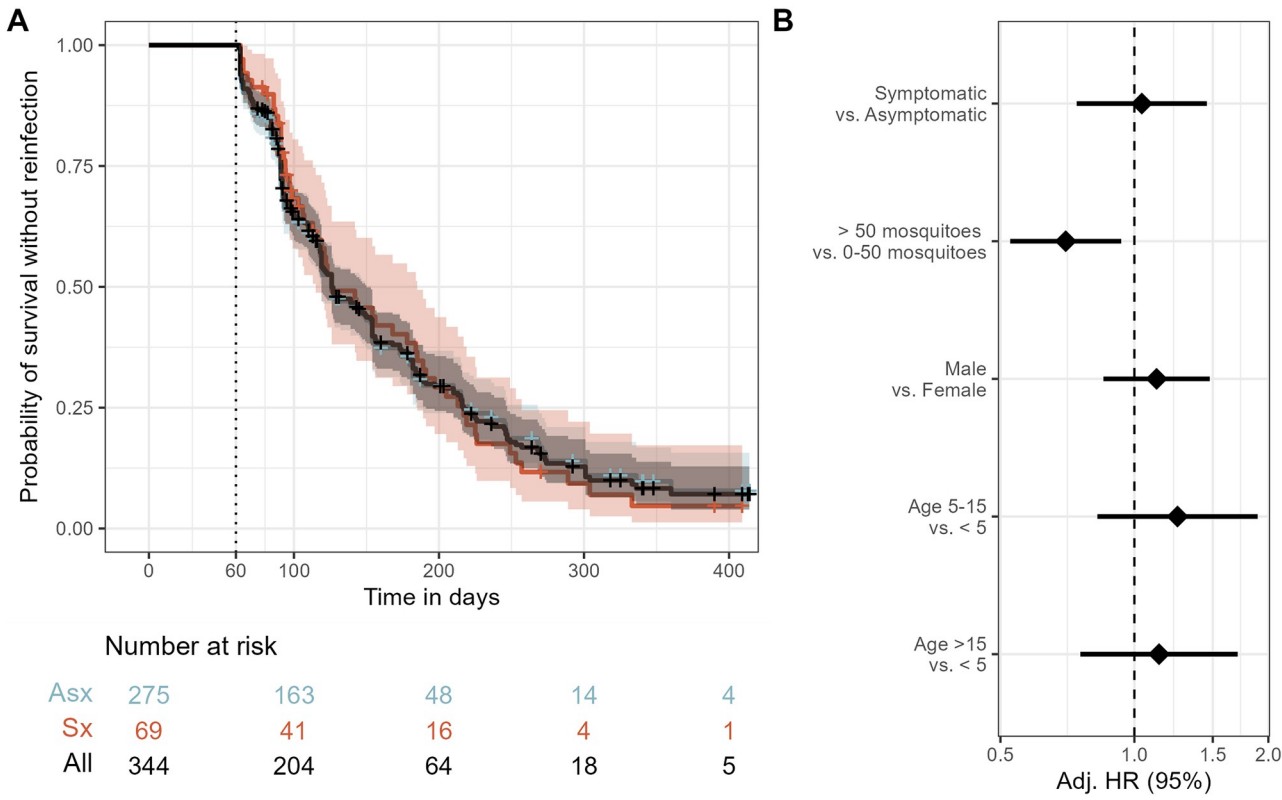

**Fig 2. Risk of reinfection overall and as a function of symptomaticity of the index infection. A)** Kaplan-Meier survival curve of time to reinfection with *P. falciparum*. The black line is the probability among all participants, and colored lines are probability stratified by index case as either asymptomatic (Asx, blue) or symptomatic (Sx, red). Note that the blue line is largely hidden by the black line. Crosses indicate infections that were not followed by a reinfection and therefore censored at the end of the study period. Vertical dotted line indicates the start of the defined time at risk of re-infection, shaded areas indicate the 95% confidence interval, and table shows the number of participants at risk stratified by symptomaticity of index infection at day 0, 100, 200, 300, and 400. **B)** Adjusted Hazard Ratios for reinfection estimated by a multivariate mixed effects Cox proportional hazards model. See methods for variable definitions. Diamonds indicate adjusted HRs, and bars show the 95% CIs. Adj. HR: Adjusted Hazard Ratio.

individual infections, the multiplicity of infection (MOI) expressed by nucleotide sequence was very closely correlated when computed based on the CSP-Th2R epitope type (ratio 1:0.95) and less so when computed by CSP DV10 (ratio 1:0.4), CSP-Th3R epitope type (ratio 1:0.6), or AMA-1 c1L type (ratio 1:0.7) (**Fig A in** S1 Text). The CSP DV10 epitope had a notably lower level of epitope type diversity relative to CSP-Th2R, CSP-Th3R, and AMA-1 c1L, with 73% (2521/3443) of the *pfcsp* haplotype occurrences detected across 861 samples coding for the major epitope type. Subsequent analyses and results for the CSP-DV10 epitope are described in S1 Text.

## Risk of reinfection with homologous CSP or AMA-1 parasites

We next investigated the risk of reinfection by homologous parasites defined by these epitope types. To do so, we partitioned *P. falciparum* infections based on the presence of specific CSP-Th2R, CSP-Th3R, and AMA-1 c1L epitope types; as a result, each participant could harbor multiple concurrent type-defined episodes with varying origin and end dates (Fig 1). By this classification, we observed 2174 CSP-Th2R, 1462 CSP-Th3R, and 1367 AMA-1 c1L type infection episodes, and the time to homologous reinfection was then calculated as the time between the end of a specific type-defined episode to the beginning of the next type-defined

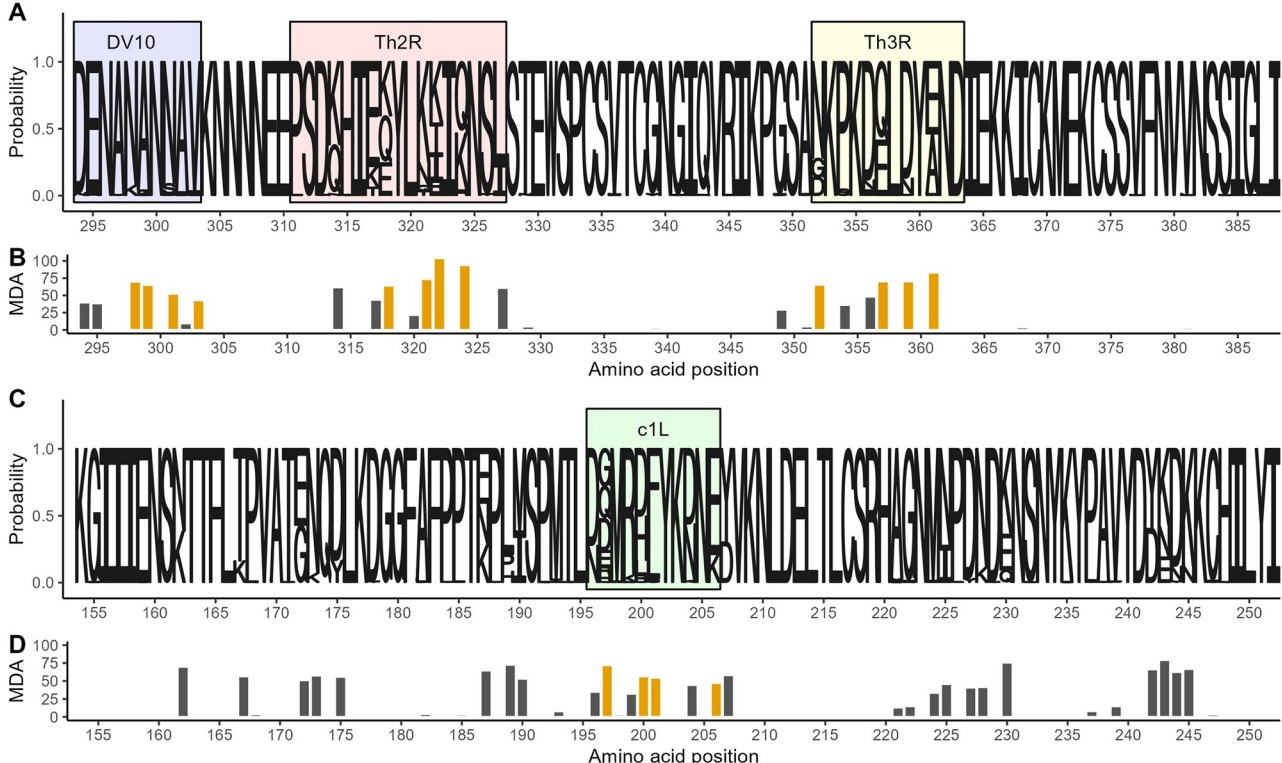

**Fig 3. Amino acid diversity of sequenced *pfcsp* and *ama1* fragments. A)** Sequence logo of the observed circumsporozoite protein (CSP) amino acid variation. The size of the letters at each position shows the probability of the specific residue. Boxes indicate the positions of known epitopes for B-cells (DV10, blue), CD4+ T-cells (Th2R, red) and CD8+ T-cells (Th3R, yellow). **B)** Result of a random forest model used to predict the whole CSP fragment nucleotide sequence using amino acid variation. Bar height indicates for each variant amino acid position the mean decrease in accuracy (MDA) of prediction. Gold indicates the four most predictive amino acid positions within each T-cell epitope. **C)** Sequence logo of the observed apical membrane antigen 1 (AMA-1) amino acid variation. The size of the letters at each position shows the probability of the specific residue. Green box marks the c1L domain. **D)** The mean decrease in accuracy for each variant amino acid position in a random forest model used to predict the whole AMA-1 fragment nucleotide sequence. Gold indicates the four most predictive amino acid positions within the c1L domain.

episode harboring the same type. The prior exclusion of index infections for which time to reinfection was less than or equal to 60 days did not result in systematic exclusion of specific epitope types (**Fig B in** S1 Text).

Using these epitope classifications of parasites, the probability of homologous reinfection within a year was 0.46 (95% CI 0.43–0.48) when classified by CSP-Th2R, 0.55 (95% CI 0.52–0.59) by CSP-Th3R, and 0.44 (95% CI 0.40–0.47) by AMA-1 c1L (**Fig C in** S1 Text). There were no differences between these risks of reinfection observed for homologous parasites and those expected for random parasite types for those classified by CSP-Th2R epitope (median Chisq 0.518, median p 0.472 log lank test), the CSP-Th3R epitope (Chisq 0.320, p 0.571 log-rank test), or AMA-1 c1L (Chisq 0.249, p 0.618 log-rank test) (**Fig C in** S1 Text). This suggests that the risk of reinfection observed for homologous parasites was not different from that expected for a heterologous parasite across both symptomatic and asymptomatic index cases.

## Risk of homologous reinfection in symptomatic and asymptomatic infections

Given the rationale that symptomatic malaria and asymptomatic infections lead to distinct immune responses to the parasite, we next compared the risk of homologous reinfection

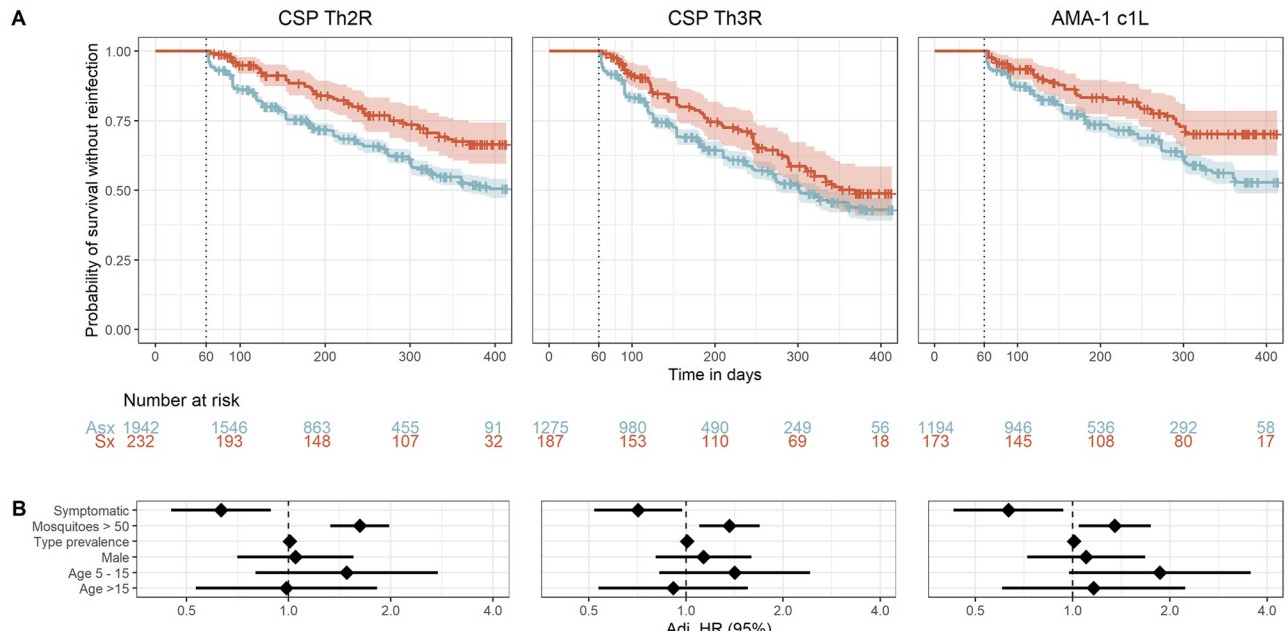

**Fig 4. Time to reinfection by homologous *P. falciparum* parasites stratified by symptomaticity of index infection. A)** Kaplan-Meier survival curves of time to reinfection by parasites with homologous CSP-Th2R, CSP-Th3R, and AMA-1 c1L type. Curves are probability stratified by index case as either asymptomatic (Asx, blue) or symptomatic (Sx, red). Crosses indicate index infections that were censored at the end of the study period. Vertical dotted line indicates the start of the defined time at risk of re-infection, shaded areas indicate the 95% confidence interval, and table shows the number of participants at risk stratified by symptomaticity of index infection at day 0, 100, 200, 300, and 400. **B)** Adjusted Hazard Ratios estimated by multivariate mixed effects Cox proportional hazards models for reinfection by parasites bearing homologous CSP-Th2R, CSP-Th3R, and AMA-1 c1L epitope types. See methods for variable definitions and reference cateogries. Diamonds indicate adjusted HRs, and bars show the 95% CIs. Adj. HR, Adjusted Hazard Ratio.

between asymptomatic and symptomatic index cases. When classified by CSP-Th2R epitope type, the probability of reinfection with a homologous parasite in 1 year was higher in asymptomatic (0.48, 95% CI 0.44–0.51) than in symptomatic (0.32, 95% CI 0.25–0.39) index cases (Fig 4). In a multivariate mixed effects Cox model, compared to asymptomatic index episodes, the presence of symptoms during the index episode was associated with a significantly reduced risk of reinfection with parasites bearing a homologous CSP-Th2R type (adjusted hazard ratio (aHR) 0.63; 95% CI 0.45–0.89; p = 0.008) (Fig 4). Similarly, the presence of symptoms was also associated with a decreased hazard of reinfection with parasites bearing homologous CSP-Th3R types (aHR 0.71; 95% CI 0.52–0.97; p = 0.033) and homologous AMA-1 c1L types (aHR 0.63; 95% CI 0.43–0.94; p = 0.022, Fig 4). The inclusion of MOI as a co-variate did not significantly impact these results for CSP-Th2R and CSP-Th3R epitopes but slightly decreased the hazard of homologous reinfection defined by AMA-1 c1L epitope types (**Table B in** S1 Text).

We then performed this comparison in randomized datasets, in which there were no consistent differences between symptomatic and asymptomatic index episodes of the risk of reinfection with homologous types defined by CSP-Th2R (log-rank (median) chisq 0.822, p 0.364), CSP-Th3R (chisq 0.495, p 0.481) or AMA-1 c1L (chisq 3.08, p 0.079), nor any association in Cox proportional hazard models between symptoms and time to reinfection (**Fig D in** S1 Text). These observations indicate that symptomatic infections are associated with a reduced risk of reinfection with parasites bearing homologous CSP or AMA-1 epitope types.

To support these findings, we also directly compared the risk of reinfection with parasites bearing homologous epitope types and random epitope types after symptomatic index episodes and separately after asymptomatic index episodes (**Fig C in** S1 Text). We observed no difference in risk of homologous and random reinfection after asymptomatic index cases for any of the CSP or AMA-1 epitope types. However, the risk of homologous reinfection was lower than the risk of random infection after a symptomatic index case when classified by CSP-Th2R epitope types (log-rank median chisq 8.945, p = 0.0028). Similar trends, though not reaching statistical significance, were observed for CSP-Th3R (log-rank median chisq 2.793 p = 0.0947) and AMA-1 c1L (log-rank median chisq 2.542, p = 0.111).

An ensemble analysis of CSP-Th2R and CSP-Th3R is presented in the S1 Text.

## Epitope type prevalence and risk of reinfection following symptomatic infection

We next investigated if delayed time to reinfection by homologous parasites following symptomatic infection was modified by the population prevalence of epitope type. To do so, we categorized the epitope types of each target into common, middling, and rare on the basis of the parasite population prevalence of the type and computed Cox proportional hazard models within each prevalence stratum. Epitopes were assigned to categories for each target to achieve approximately equal population frequency of each category (Fig 5). To varying degrees, we observed reduced hazards of homologous reinfection for each epitope type within each prevalence stratum, with the lowest hazards for homologous reinfection associated with symptomatic malaria observed for the rare CSP-Th2R (aHR 0.35, 95% CI 0.16–0.77, p = 0.010) and AMA-1 c1L (aHR 0.23, 95% CI 0.08–0.68, p = 0.008) epitope types. There was moderate support for effect modification by epitope type frequency for models of both the CSP-Th2R (p = 0.096 by log likelihood test) and AMA-1 c1L (p = 0.055) epitope types. These observations suggest that epitope type rarity within a parasite population modifies the effect of symptoms on the risk of homologous reinfection, in that symptomatic malaria more strongly reduces the risk of homologous reinfection for the rarest epitope types.

## Discussion

In this longitudinal cohort study with 14 months of follow-up in a high transmission setting in western Kenya, we investigated factors associated with the time to reinfection with malaria parasites bearing homologous CSP and AMA-1 sequences. We observed that, compared to asymptomatic infections, symptomatic malaria was associated with a reduced risk of reinfection by parasites bearing identical epitope types. Additionally, this increased protection from homologous reinfection that was associated with symptomatic malaria was observed across the full range of diverse CSP-Th2R, CSP-Th3R, and AMA-1 c1L types but was greatest for the rarest types. Given that delay in reinfection with homologous parasites is a signature of acquired functional immunity, these observations suggest that, compared to asymptomatic infections that are common in high-transmission settings, symptomatic infections more effectively inscribe anti-parasite immunity to build protection against malaria.

Our observation that the presence of symptoms is associated with a decreased risk of reinfection with parasites bearing homologous epitope types is consistent with enhanced engagement of the immune system during symptomatic infections. Our CSP types were defined by polymorphic residues in the known CD4+ [16] and CD8+ T-cell [17] epitopes in the C-terminal domain, suggesting that T-cell responses are primed more effectively by symptomatic infections. T-cells participate directly and indirectly in functional immunity to malaria (reviewed in [19]), particularly directed towards the pre-erythrocytic stages of the parasites

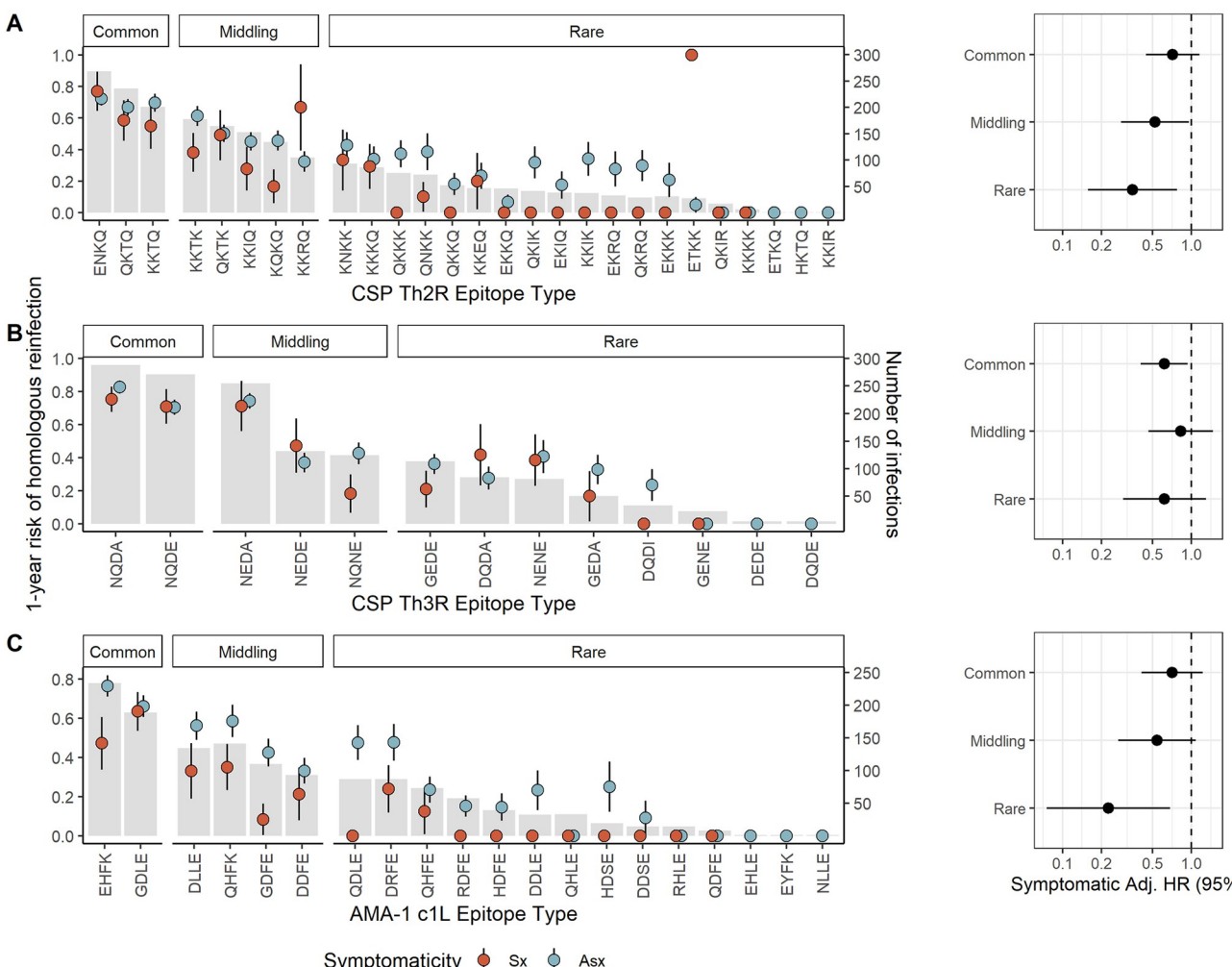

**Fig 5. Prevalence of epitope types and time to homologous reinfection following asymptomatic and symptomatic infections. Left**: Gray bars indicate the number of infections harboring each CSP-Th2R (**A**), CSP-Th3R (**B**), or AMA-1 c1L (**C**) epitope type (right y-axis). Dots indicate the proportion of each index infection harboring the epitope type that was followed by homologous reinfection after 1 year (left y-axis) for either symptomatic (red) or asymptomatic (blue) index infections, with lines as standard error. **Right**: Adjusted Hazard Ratios for symptomatic index infections estimated by multivariate mixed effects Cox proportional hazards models for reinfection by parasites bearing homologous CSP-Th2R epitopes (A), CSP-Th3R (B), or AMA-1 c1L (C) epitopes that are individually classified as common, middling, and rare, as indicated in the bar plot. Hazard ratios adjusted for age, epitope type prevalence, gender, and transmission season. Circles indicate adjusted HRs, and bars show the 95% CIs. Adj. HR, Adjusted Hazard Ratio.

that express CSP. Recognition of these CSP epitopes by CD8+ [20] or CD4+ [21] T-cells has been associated with protection from malaria, although, notably, we did not observe overall protection from reinfection with homologous compared to random parasites (**Fig C in** S1 Text), suggesting that these functional responses to CSP epitopes are either promoted during symptomatic malaria or attenuated during an asymptomatic infection. Similarly, AMA-1 types were defined on residues comprising the c1L domain that encode the main targets of naturally-acquired anti-AMA-1 immunity, [18,22] supporting the notion that symptomaticity reflects the development of functional immunity. Consistent with this interpretation, in a cohort of Malian children, [23] febrile malaria resulted in the upregulation of multiple markers of adaptive immunity, including T-cell stimulation, opsonic and non-opsonic phagocytosis, and antigen processing. Our findings extend these observations to suggest that

such immune effectors are strain-specific and ineffectively promoted by asymptomatic infections.

We observed that the risk of homologous reinfection was reduced following symptomatic infections across a diversity of CSP-Th2R, CSP-Th3R, and AMA-1 c1L types with widely varying population frequencies (Fig 5). The otherwise broad protection following symptomatic infection suggests that this phenomenon is not a biological consequence of specific sequence types or an epidemiologic artifact driven only by rare epitopes which are necessarily unlikely to recur. Interestingly, though risk reductions were similar across prevalence categories for parasites defined by CSP-Th3R type, we observed modification of this effect by the prevalences of CSP-Th2R and AMA-1 c1L epitope types, with a greater protection following symptomatic infection among the rarest types than the most common types. This suggests that the rarity of these epitope types may be a consequence of the protection conferred by symptomatic infections harboring them, and we hypothesize that anti-parasite immunity may shape parasite diversity. Alternatively, compared to common epitopes, these rare epitopes may be more likely to cause symptomatic episodes owing to a lower background of exposure across the population.

Several facts support the notion that the observed delay in homologous reinfection following symptomatic malaria represents a legible signature of functional anti-parasite immunity. Firstly, a delayed time-to-reinfection with a homologous pathogen strain is a common phenotype of functional immunity. Secondly, this delay in time-to-reinfection was observed when parasites were classified on the basis of known targets in CSP [16,17] and AMA-1 [18,24] of functional responses that have been measured and correlated with protection in a wide variety of approaches. Thirdly, the effect was observed across a diverse range of sequences and epitope type prevalences, indicating that it was not artifactually driven by specific over-represented epitopes. Additionally, and in contrast, the delay was not present when comparing time-to-reinfection with random epitope types (**Fig D in** S1 Text), indicating that the effect was specific to parasites harboring an exact epitope match and not an effect generalizable to alternate epitopes of similar population prevalence. Further, when the time to reinfection with parasites bearing homologous epitope types was directly compared to time to random reinfection, homologous reinfection was delayed relative to random reinfection only after symptomatic exposure (**Fig C in** S1 Text). Finally, this effect was not present when analyzed agnostic to parasite genotype (Fig 2), reflecting the importance of CSP and AMA-1 epitope type. Though this phenomenon is not easily measurable in most study designs, future longitudinal molecular epidemiologic studies could screen a wide variety of known and putative parasite antigens and correlate variants with the degree of delay in reinfection following a symptomatic episode as an approach to identifying promising targets of natural immunity.

It is notable that this phenomenon was observed for parasites classified by sequences of antigens expressed during both the pre-erythrocytic and the blood stage. Given that AMA-1 is expressed during the blood stage, the rationale is straightforward for symptomaticity during the blood stage to reflect enhancement of functional immunity directed against AMA-1. However, how might symptomaticity during the blood stage mediate functional immunity to variants of the CSP antigen that is expressed during the pre-erythrocytic stage? CSP is expressed on sporozoites and during the liver stage but not during the blood stage, and the typical delay between release from the liver and onset of symptoms of at least 4–8 days (or longer in semi-immune individuals) suggests CSP should no longer be present and eliciting responses. One potential explanation is persistence of CSP in exo-erythrocytic parasite forms and antigen-presenting cells, which can be found in the skin, liver, and lungs for an undetermined period of time. [25] However, there is clear interplay between blood- and liver-stage infection and immunity, as existing blood stage infection disrupts the immune response to concurrent pre-

erythrocytic infection or vaccination with attenuated sporozoites. [26,27] Relevant to our findings is the evidence that exposure to pre-eryothrocitic parasites modulates the response and severity of blood stage infections. For example, liver stage activation of distinct γδ T cell subsets protects against severe disease during the blood stage in mouse models. [28] Additionally, given their critical role in pre-erythrocytic immunity, [29,30] effector CD8+ T-cells primed by CSP exposure may be converted more efficiently by the mechanisms which produce symptoms into memory CD8+ cells with the ability to rapidly confer pre-erythrocytic protection on re-exposure. [31] Finally, symptomaticity may cause or result from processes that influence regulatory T-cells, which have cryptic roles in adaptive immunity to malaria but are generally thought to modulate inflammation to parasitemia and acquisition of functional immunity. [32]

Two other observations are notable. First, the degree of protection against homologous reinfection associated with symptomatic infection was fairly consistent across classifications by DV10 (aHR 0.75), CSP-Th2R (aHR 0.63), CSP-Th3R (aHR 0.71), combined CSP-Th2R/Th3R (aHR 0.56), and AMA-1 c1L (aHR 0.63) epitope types. This is despite the varied effector mechanisms that are associated with each epitope, including recognition by B-cells, CD4+ [16] or CD8+ [17] T-cells, and by polyclonal sera. [18] This shared degree of protection suggests that symptomaticity may promote the acquisition of effective immunity to similar degrees across different effector mechanisms. Secondly, despite the vast *pfcsp* sequence diversity, the 155 unique *pfcsp* nucleotide haplotypes reduced to just 27 CSP-Th2R types with little loss in within-host diversity: within individuals, the observed number of CSP-Th2R epitope types was correlated nearly one-to-one with the number of unique *pfcsp* haplotypes, demonstrating a low level of redundant or silent nucleotide substitutions at the Th2R epitope. By contrast, significant loss of within-host diversity was observed when *pfcsp* sequences were classified by CSP-DV10 and CSP-Th3R epitopes, which reduced to 8 and 14 types, respectively (**Fig A in** S1 Text). This suggests that, within polyclonal infections, amino acid diversity of CSP is maximized at the Th2R epitope.

This study has some limitations. Precisely defining the start and end of infections is challenging. We mitigated this challenge by using the longitudinal study design and classifying the beginning and end of infections based on prior and subsequent samples being *P. falciparum*-negative to avoid misclassifying persistent infections. Additionally, symptomatic infections were necessarily treated upon diagnosis, which could bias comparisons with asymptomatic infections which ended without treatment. Thus, differences in the risk of homologous reinfections may in fact reflect the impact of treatment rather than the impact of epitope type-specific immunity. However, we did not observe a difference in risk of overall reinfection between symptomatic and asymptomatic cases (Fig 2), nor was there a difference in risk of reinfection with random epitope types (**Fig D in** S1 Text). Thirdly, our genotyping approach is subject to false discovery. This was mitigated by the application of stringent read- and haplotype-quality filtering. Finally, an important limitation of this study is that it does not explore the protection conferred by asymptomatic infection relative to the absence of an index infection. Asymptomatic infections likely result from prior establishment of protective immune responses that control parasite density and/or disease, [33–35] and they also can lead to humoral immune responses and protection. [36–38] Our observations are not incompatible with existing insights into the interplay between asymptomatic infection and immunity, but rather provide evidence that symptomatic episodes lead to more durable protection against homologous reinfection than do asymptomatic infections.

In our longitudinal molecular epidemiologic study, compared to asymptomatic *P. falciparum* infections, symptomatic malaria was associated with an increased time to reinfection by parasites harboring homologous CSP and AMA-1 epitope types. The fact that these

epitope types were defined by variant residues in known immunogenic regions supports the notion that a differential time-to-reinfection with homologous parasite strains between asymptomatic infection and symptomatic malaria represents a legible signature of functional immunity to *P. falciparum*. Future studies can adapt this framework to identify putative targets of naturally-acquired immunity in re-infection cohorts and identify additional parasitologic, clinical, or host factors that modify the process by which anti-disease immunity is inscribed.

## Materials and methods

### Inclusion and ethics statement

Written informed consent was provided by all adults and by parents or legal guardians for individuals under 18 years of age. Additional verbal assent was given by individuals between 8 and 18 years of age. The study was approved by the ethical review boards of Moi University (2017/36) and Duke University (Pro00082000).

### Study design and sampling

The longitudinal cohort consisting of participants in three villages in Western Kenya has been described previously. [13,39] In brief, all household members over 1 year of age were offered enrollment, and demographic data was collected for each participant. Dried blood spot (DBS) samples were collected from participants every month and any time a participant experienced symptoms associated with malaria, when they additionally were tested for malaria with a rapid diagnostic test and treated if positive with Artemether-Lumefantrine. We analyzed parasite gene sequences that have been previously reported. [13,40] Briefly, *P. falciparum* was detected in genomic DNA isolated from all DBS using a real-time PCR assay, [41] variant targets in the genes encoding the apical membrane antigen-1 (*pfama1*) protein and the circumsporozoite protein (*pfcsp*) were amplified, these were sequenced on Illumina MiSeq, and sequencing reads were analyzed and haplotypes filtered as described previously. [13] The *pfama* and *pfcsp* gene fragments were translated into protein sequences. Seven CSP and three AMA-1 nucleotide sequences containing stop-codons were removed. Amino acid sequence logos were made across all the sequences of either CSP or AMA-1 based on the probability of amino acid residues at each position.

### Random forest models

Random forest algorithms were generated to predict the nucleotide sequence based on all the variant amino acid positions. The number of amino acid positions to try per node split (mtry) was first evaluated using the TuneRF function where random forest models of 100 trees were created with varying mtrys on the CSP sequences. A final mtry of 10 was selected as this was the first mtry that resulted in on out-of-bag (OOB) error rate below 0.1. Each final random forest model contained 2000 trees, and the OOB error rate was evaluated. The mean decrease accuracy for each amino acid position was used as a measure of the importance of each position.

### Exposure classifications

A sample was defined as an asymptomatic malaria case when positive for *P. falciparum* by qPCR in the absence of symptoms. A sample was defined as a symptomatic malaria case when positive for *P. falciparum* by rapid diagnostic test (RDT) and qPCR in the presence of a

symptom consistent with malaria (aches, chills, congestion, cough, diarrhea, fever, or vomiting) during a sick visit.

Samples from each participant were classified based on the result of *P. falciparum* qPCR for the purpose of defining episodes of *P. falciparum* infection (Fig 1). For successive *P. falciparum* positive samples, the first positive sample was classified as the start of an infection episode, and the last positive sample before a negative sample was classified as the end of the infection episode. Symptomatic infections were classified as the end of an episode, since they were treated. All infections within 14 days of treatment were excluded from the dataset owing to the possibility of residual detectability following effective treatment. These samples at the end of an episode were treated as index cases, and time to reinfection was then calculated as the time between index case and the next *P. falciparum* positive sample. Index cases not followed by a *P. falciparum* positive sample were censored at the end of the study.

Parasite epitope types detected at each time point were the basis to re-classify episodes of *P. falciparum* infection into episodes of parasite epitope-type infection (Fig 1). Similar to above, we defined the beginning of each epitope episode as the first sample in which it was detected following a sample in which it was absent, and the end as the last sample in which it was detected prior to a sample in which it was absent. Censoring of infections within 14 days of a symptomatic event was performed independently of epitope types. Because most infections harbored more than one epitope, each infection episode resulted in more than one epitope-type episode.

## Computing time to re-infection

Re-infection was defined as the re-appearance of any *P. falciparum* infection in a person who cleared a prior infection. A homologous re-infection was defined as the re-appearance in the same person of a parasite harboring an identical epitope type to that observed previously in that person. The time to reinfection was calculated as the interval between the end of the prior episode and the beginning of the next episode. These intervals were computed for a person's individual infection episodes, and then within these infection episodes for a person's epitope type episodes.

To determine whether our results were different from those expected by chance, we also determined time to homologous reinfection for CSP-Th2R, CSP-Th3R, and AMA-1 c1L in randomized datasets. Epitope types were randomized by permuting at the sample level within epitope MOI categories: mono-epitope (1), pauci-epitope (2–5), and multi-epitope (> 5) type episodes. This permutation scheme not only preserved the epitope type prevalences, MOI distribution, and sampling structure of the original dataset, but also maintained identical (mono) or similar (pauci- and multi-) MOIs for individual epitope-type infection episodes, reflecting a relevant null at the individual sample level and for the overall dataset. The time to homologous reinfection in the permuted dataset was then calculated as the interval between episodes with the same type as described above. Epitope types were permuted 1000x, generating 1000 randomized datasets for both CSP and AMA-1. Subsequent time-to-event analyses were performed on all permuted datasets as detailed below, and all reported measures are medians of these 1000 randomized datasets.

## Hazard of reinfection

Kaplan-Meier curves were fit to estimate the proportion of individuals reinfected 1 year after an index infection. The proportional hazards assumption was assessed using Kaplan-Meier curves and plots of the log-hazard over time and, owing to the sampling scheme, only held for events with time to reinfection greater than 60 days. Thus, all subsequent analyses were

performed on the subset of index infections for which time to reinfection was greater than 60 days.

Mixed-effect Cox regression models were used to estimate the hazard of reinfection over the entire study period (Eq 1):

$$\frac{h_1(t)_i}{h_0(t)_i} = \exp(\alpha_i + \beta_1 \text{Symptomatic infection}_i + \beta_2 \text{Age 5to15}_i + \beta_3 \text{Age over 15}_i$$

$$+ \beta_4 \text{Male}_i + \beta_5 \text{Transmission intensity} + \epsilon_i)$$

These models include as covariates age (categorized as $<5$, 5–15, and $> 15$ years), sex, transmission intensity ($\leq 50$ or $> 50$ mosquitoes collected in the prior 14 days across all study sites), and a random intercept at the participant level ($\alpha_i$) to account for repeated measures and correlated outcomes within individuals. A log-normal distribution was used for the random effect. $\epsilon$i represents the error term of the model.

The hazard of homologous reinfection and hazard of random sub-type reinfection across participants through the entire study period was calculated using multivariate mixed effects cox regression as above with the addition of sub-type prevalence as a co-variate (Eq 2):

$$\frac{h_1(t)_i}{h_0(t)_i} = \exp(\alpha_i + \beta_1 \text{Symptomatic infection}_i + \beta_2 \text{Age 5 to 15}_i + \beta_3 \text{Age over 15}_i$$

$$+ \beta_4 \text{Male}_i + \beta_5 \text{Type prevalence} + \beta_6 \text{Transmission intensity} + \epsilon_i)$$

We additionally evaluated whether adding MOI as a co-variate to the multivariate mixed effects cox models influenced the hazard of homologous reinfection during the study period. In this case, MOI was defined as the number of distinct epitope types observed in the final sample of an epitope-type infection episode for a given epitope. Models including MOI as a continuous measure and as a categorical measure (1, 2–5, or >5) were evaluated and compared to the main models (Eq 2) based on model point estimates and confidence intervals (adjusted Hazard Ratios), likelihood-ratio tests, Akaike information criteria (AIC) values.

We assessed effect measure modification by epitope type prevalence by comparing the 95% confidence intervals for adjusted stratum-specific estimates and by including an interaction term between prevalence and sympomaticity in the model. The prevalence of an epitope type was defined as the proportion of all epitope-type episodes that harbored the epitope type. We used a likelihood ratio test to statistically compare model fit with and without (Eq 2) the interaction term.

The models for time to homologous reinfection and time to random reinfection were compared using a log-rank test. Statistical significance was assessed at an α level of 0.05. All statistical analysis and visualization were done using R (4.1.0) [42] with the packages tidyverse (1.3.1), [43] Biostrings (2.62.0), [44] ggseqlogo (0.1), [45] RandomForest(4.6–14), [46] survival (3.2–13), [47] survminer (0.4.9), [48] coxme (2.2–16), [49] ggfortify (0.4.13), [50] ggpmisc (0.4.4), [51] lubridate (1.8.0), [52] ggalt (0.4.0), [53] ggpubr (0.4.0). [54]

## Supporting information

**S1 Text. Supplemental results.** Table A. Comparison of index infection episodes included and excluded from survival analyses. Table B. Cox mixed effect model estimates for reinfection with parasites bearing homologous epitope types after symptomatic and asymptomatic exposure, without and with adjustment for MOI. Fig A. Comparative diversity of nucleotide haplotypes and epitope types. Fig B. Comparison of epitope type frequencies in index infections with time to reinfection less than or greater than 60 days. Fig C. Risk of

reinfection with *P. falciparum* parasites bearing homologous epitopes. Fig D. Time to reinfection with random epitope type stratified by symptomaticity. Fig E. Analyses of parasites defined by CSP DV10. Fig F. Analyses of parasites defined by combined CSP-Th2R/Th3R epitopes.
(DOCX)

## Acknowledgments

We appreciate the study implementation skills of project manager and field technicians in Webuye and Eldoret: J. Kipkoech Kirui, I. Khaoya, L. Marango, E. Mukeli, E. Nalianya, J. Namae, L. Nukewa, E. Wamalwa, and A. Wekesa. We thank A. Nantume, J. Saelens, and Z. Lapp (each of Duke University) for their help with laboratory samples, data processing, and helpful discussion. Ultimately, we are indebted to the people who participated in our study.

## Author Contributions

**Conceptualization:** Jens E. V. Petersen, Steve M. Taylor.

**Formal analysis:** Christine F. Markwalter, Jens E. V. Petersen, Erica E. Zeno, Kelsey M. Sumner.

**Funding acquisition:** Andrew A. Obala, Wendy Prudhomme-O'Meara, Steve M. Taylor.

**Investigation:** Christine F. Markwalter, Jens E. V. Petersen, Elizabeth Freedman, Judith N. Mangeni, Lucy Abel, Andrew A. Obala, Wendy Prudhomme-O'Meara, Steve M. Taylor.

**Methodology:** Christine F. Markwalter, Jens E. V. Petersen, Erica E. Zeno, Elizabeth Freedman, Judith N. Mangeni, Wendy Prudhomme-O'Meara, Steve M. Taylor.

**Project administration:** Andrew A. Obala, Wendy Prudhomme-O'Meara, Steve M. Taylor.

**Resources:** Kelsey M. Sumner, Lucy Abel, Andrew A. Obala, Steve M. Taylor.

**Software:** Kelsey M. Sumner.

**Supervision:** Andrew A. Obala, Wendy Prudhomme-O'Meara, Steve M. Taylor.

**Validation:** Christine F. Markwalter.

**Visualization:** Christine F. Markwalter.

**Writing – original draft:** Jens E. V. Petersen.

**Writing – review & editing:** Christine F. Markwalter, Erica E. Zeno, Elizabeth Freedman, Judith N. Mangeni, Lucy Abel, Andrew A. Obala, Wendy Prudhomme-O'Meara, Steve M. Taylor.

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
