## [Decision Letter · Decision Letter 0]

20 Mar 2023

Dear Dr. Taylor,

Thank you very much for submitting your manuscript "Symptomatic malaria enhances protection from reinfection with homologous Plasmodium falciparum parasites" for consideration at PLOS Pathogens. As with all papers reviewed by the journal, your manuscript was reviewed by members of the editorial board and by several independent reviewers. In light of the reviews (below this email), we would like to invite the resubmission of a significantly-revised version that takes into account the reviewers' comments.

The main observation made in this study is of potential interest. However, careful attention should be paid to how the study has been framed in the Introduction, how results have been interpreted, and whether possible limitations and alternative explanations have been thoroughly addressed. In particular, in comparing symptomatic and asymptomatic infections, the authors should consider more deeply what other confounders might be at play, and whether new analyses as suggested by Reviewers will serve to clarify.

We cannot make any decision about publication until we have seen the revised manuscript and your response to the reviewers' comments. Your revised manuscript is also likely to be sent to reviewers for further evaluation.

Sincerely,

Ashraful Haque

Academic Editor

PLOS Pathogens

P'ng Loke

Section Editor

PLOS Pathogens

Kasturi Haldar

Editor-in-Chief

PLOS Pathogens

orcid.org/0000-0001-5065-158X

Michael Malim

Editor-in-Chief

PLOS Pathogens

orcid.org/0000-0002-7699-2064

The main observation made in this study is of potential interest. However, careful attention should be paid to how the study has been framed in the Introduction, how results have been interpreted, and whether possible limitations and alternative explanations have been thoroughly addressed. In particular, in comparing symptomatic and asymptomatic infections, the authors should consider more deeply what other confounders might be at play, and whether new analyses as suggested by Reviewers will serve to clarify.

Reviewer's Responses to Questions

**Part I - Summary**

Reviewer #1: Markwalter and colleagues present an interesting secondary analysis of previously collected longitudinal cohort. Parasite infections were classified based on CSP and AMA1 variants, and as symptomatic and asymptomatic infections. Authors suggest that symptomatic infections have an increased time to reinfection with a homologous parasite strain.

Commenting on the modelling used in this manuscript is outside of my expertise. However, I have some large concerns about the framing and interpretation of the data. I suggest that authors have a careful think about this interpretation in revision.

Reviewer #2: In this very well-written paper by Markwalter et al, the authors investigated factors associated with the time to reinfection with malaria parasites bearing homologous CSP and AMA-1 sequences. To perform these analyses, the authors performed a secondary analysis of already published data (Sumner et al, Nature Comms, 2021) where they sequenced pfcsp and pfama from 861 and 724 samples, respectively, among asymptomatic (n=902) and symptomatic (n=174) Pf infections. To enable an analysis of reinfection with homologous parasites, they assessed parasite genotypes, and classified genotypes into the most informative variable AA positions that occurred in previously described, immunologically-relevant antigen segments. The authors excluded reinfections that occurred within 60 days due to methodologic issues with sampling asymptomatic infections every 30 days, but included sensitivity analyses to compare these excluded infections with those that were included and did not note significant differences between the two. When comparing the risk of reinfection between symptomatic and asymptomatic index infections irrespective of parasite sequence type, they found that presence of symptoms during the index infection was not associated with the hazard of reinfection. However, when considering parasite sequence, the authors found that the hazard of reinfection with a homologous parasite was ~30-40% lower following symptomatic infections compared with asymptomatic infections. This is an intriguing result, and suggests that, in contrast to asymptomatic infections, symptomatic infections may be associated with acquisition of a strain-specific, anti-parasite immune response. In all, I really enjoyed reading this manuscript. The methods and analysis were very clearly described, and the results and message of the paper easy to follow.

**Part II – Major Issues: Key Experiments Required for Acceptance**

Reviewer #1: Authors in the introduction frame the data by suggesting that symptomatic malaria infection is associated with induction of more robust immunity compared to asymptomatic infections. However, I believe that causality has been miss-interpreted in this framing. While the literature does show that symptomatic infection is associated with high levels of inflammatory responses, this should not be interpreted as asymptomatic infections being ‘low immune induces’. Indeed, the presences of an asymptomatic infection is likely due to the already existing, prior establishment of protective immune responses. This prior immunity is then able to control parasite density and/or limit immune pathology. Consistent with protective prior immunity in asymptomatic cases in the authors data set is the significantly lower parasite burden in asymptomatic infections.

While there is some literature in other infections and vaccinations that symptoms can be associated with induction of protective immunity to the best of my knowledge evidence for this in malaria is lacking. At the very least, the authors do not provide sufficient citation of prior literature that would support this hypothesis in malaria in the introduction.

Some specific examples that need to be reconsidered:

Introduction line 73-75 – these statements sound like the authors are proposing that symptomatic malaria results in better immunity induction compared to asymptomatic infection. However Ref 15 Portugal et al, doenst support this concept, and only looks at symptomatic malaria, not compared to asymptomatic malaria in comparison. This citation is also used to say that febrile malaria resulted in upregulation of multiple markers of adaptive immunity – which is true, but that doesn’t mean that asymptotic infection doesn’t result in increased immune development also

Statement line 174: Presence of symptoms during an index infection reflects of immune response to the parasite, we next…

A symptomatic infection, with high parasite infection is indicative of uncontrolled parasite growth, and while inflammation is indicative of an active immune response, that is not directly linked to the development of a protective adaptive response – aka inflammation doesn’t necessarily mean robust development of protection. If the authors know of data that does support this hypothesis, these papers need to be specifically cited. I interpret the existing data in the opposite direction, an asymptotic infection is low density, indicating a adaptive response that exists already and can control the parasite. Authors need to consider whether there is any existing data from the malaria immunity space that supports their framing and hypothesis sufficiently. For example, there is data on the other side – that parasite density doesn’t associated with antibody development (see Chan et al, CRM, 2020), and that antibodies are induced in asymptomatic infection (for example in Pv, but similar concept Longley, Malaria Journal, 2017).

Along with the framing/hypothesis underpinning the paper, the discussion and interpretation of the results needs to be carefully re-considered.

Specifically, the following:

Authors do not discuss evidence that symptomatic blood stage infection disrupts immunity to pre-erythrocytic stages. See lines 244 and section 289-302

Authors need to discuss the data of

doi: 10.1016/j.celrep.2016.11.060

DOI: 10.1371/journal.ppat.1009594

Both of these studies clearly show that blood stage infection (where symptoms occur) disrupt immunity to the sporozoite stage (CSP). As such, the hypothesis/rational for the symptomatic impact of improving immunity is not at all supported by these studies.

Line 262 – the greatest protection following symptomatic infection among the rarest types than the most common types. This suggests that the rarity of these epitope types may be a consequence of the protection conferred by symptomatic infections harbouring them, and highlights a mechanisms by which anti-parasite immunity shapes parasite diversity. Again, authors need to carefully re-consider causality in their framing and interpretation of the data. Asymptomatic infection may here be due to prior exposure and pre-existing immunity to existing variants.

Reviewer #2: I have one major suggestion with the analysis. 1) Asymptomatic infections are more likely to be persistent, and a higher detection of homologous reinfections following asymptomatic infections could represent misclassification of persistent infections that transiently escape the limit of detection as new infections. Although the authors do note this in the limitations section of the discussion, could the authors perform an additional sensitivity analysis to define the end of infection as the absence of the parasite or type in the subsequent 2 visits? This would help with interpretation of the results, in my opinion.

My other concern was also addressed in the limitations section of the discussion: 1) a main difference between symptomatic and asymptomatic infections clinically is that symptomatic infections are treated, whereas asymptomatic infections are not; differences in the risk of reinfection by homologous infections may in fact reflect the impact of treatment, rather than an impact of immunity. (suggest adding something along those lines in the limitations)

**Part III – Minor Issues: Editorial and Data Presentation Modifications**

Reviewer #1: (No Response)

Reviewer #2: Minor:

Text lines 114-118 and Figure 2. The authors show that presence of symptoms was not associated with risk of reinfection irrespective of sequence. Interestingly, having >50 mosquitos though appears to be counterintuitively associated with a significantly lower risk of reinfection (Fig 2B). The authors should provide some discussion of these results

Figure 2B: Please define reference groups for multivariate mixed effects model (specifically, define the reference group for >50 mosquitos – and is this a human biting rate? would help reader as opposed to having them look back at the methods) Also, in the methods this states “mosquitoes collected in the prior 14 days across study site”. Does that mean that this is not a marker of household level transmission but more a marker of village level transmission? Would specify (and cite a reference in methods; apologies if I missed that)

Line 181: “with parasites bearing…”

Line 196: “the risk of reinfection”

PLOS authors have the option to publish the peer review history of their article (what does this mean?). If published, this will include your full peer review and any attached files.

Reviewer #1: No

Reviewer #2: No
---

## [Decision Letter · Decision Letter 1]

24 May 2023

Dear Dr. Taylor,

We are pleased to inform you that your manuscript 'Symptomatic malaria enhances protection from reinfection with homologous Plasmodium falciparum parasites' has been provisionally accepted for publication in PLOS Pathogens.

Best regards,

Ashraful Haque

Academic Editor

PLOS Pathogens

P'ng Loke

Section Editor

PLOS Pathogens

Kasturi Haldar

Editor-in-Chief

PLOS Pathogens

orcid.org/0000-0001-5065-158X

Michael Malim

Editor-in-Chief

PLOS Pathogens

orcid.org/0000-0002-7699-2064

Reviewer Comments (if any, and for reference):

Reviewer's Responses to Questions

**Part I - Summary**

Reviewer #1: All comments have been adequately addressed.

Reviewer #2: The authors have adequately addressed my comments and the revised manuscript has been strengthened

**Part II – Major Issues: Key Experiments Required for Acceptance**

Reviewer #1: All comments have been adequately addressed.

Reviewer #2: n/a

**Part III – Minor Issues: Editorial and Data Presentation Modifications**

Reviewer #1: All comments have been adequately addressed.

Reviewer #2: n/a

PLOS authors have the option to publish the peer review history of their article (what does this mean?). If published, this will include your full peer review and any attached files.

Reviewer #1: No

Reviewer #2: No

---

## [Editor Report · Acceptance letter]

8 Jun 2023

Dear Dr. Taylor,

We are delighted to inform you that your manuscript, "Symptomatic malaria enhances protection from reinfection with homologous Plasmodium falciparum parasites," has been formally accepted for publication in PLOS Pathogens.

Best regards,

Kasturi Haldar

Editor-in-Chief

PLOS Pathogens

orcid.org/0000-0001-5065-158X

Michael Malim

Editor-in-Chief

PLOS Pathogens

orcid.org/0000-0002-7699-2064